# A MULTI-TASK U-NET FOR SEGMENTATION WITH LAZY LABELS

## ABSTRACT

The need for labour intensive pixel-wise annotation is a major limitation of many fully supervised learning methods for image segmentation. In this paper, we propose a deep convolutional neural network for multi-class segmentation that circumvents this problem by being trainable on coarse data labels combined with only a very small number of images with pixel-wise annotations. We call this new labelling strategy 'lazy' labels. Image segmentation is then stratified into three connected tasks: rough detection of class instances, separation of wrongly connected objects without a clear boundary, and pixel-wise segmentation to find the accurate boundaries of each object. These problems are integrated into a multi-task learning framework and the model is trained end-to-end in a semi-supervised fashion. The method is demonstrated on two segmentation datasets, including food microscopy images and histology images of tissues respectively. We show that the model gives accurate segmentation results even if exact boundary labels are missing for a majority of the annotated data. This allows more flexibility and efficiency for training deep neural networks that are data hungry in a practical setting where manual annotation is expensive, by collecting more lazy (rough) annotations than precisely segmented images.

## 1 INTRODUCTION

Image segmentation has been an active research field in the past decades. Deep learning approaches play an increasingly important role and have become state-of-the-art in various segmentation tasks (Huang et al., 2018; Khoreva et al., 2017; Tsutsui et al., 2018; Ghosh et al., 2018; Litjens et al., 2017). Though fully supervised segmentation neural networks have a shown great success, one of their most challenging issues is the need for pixel-level annotations to train them. Obtaining such annotations usually requires a great amount of manual work and is therefore expensive.

In this paper, we propose a multi-class and multi-instance segmentation approach that we split into three relevant tasks: detection, separation and segmentation (cf. Figure 1). Doing so, we obtain a semi-supervised learning approach that is trained with so-called "lazy" labels, that is a lot of coarse annotations of class instances together with only a few pixel-wise annotated images that can be obtained from the coarse labels in a semi-automated way. In the following, we will refer to weak (resp. strong) annotations for coarse (resp. accurate) labels and denote them as WL and SL.

**Task 1** detects and classifies each object and roughly determines its region through an under-segmentation mask. Instance counting can be obtained as a by-product of this task. As the main objective is instance detection, exact labels for the whole object or its boundary are not necessary at this stage. We use instead weakly annotated images in which a rough region inside each object is marked, cf. the most top left part of Figure 1. For segmentation problems with a dense population of instances, such as the food components (see e.g., Figure 1), cells (Guerrero-Pena et al., 2018; Ronneberger et al., 2015), glandular tissue, or people in a crowd (Wang et al., 2018b), separating objects sharing a common boundary is a well known challenge. We can optionally perform a second task (**Task 2**) that focuses on the separation of instances that are connected without a clear boundary dividing them. Also for this task we rely on WL to reduce the burden of manual annotations: touching interfaces are specified with rough scribbles, cf. top left part of Figure 1. **Task 3** finally tackles the pixel-wise classification of the instances. It requires strong annotations that are accurate up to the boundaries of the objects. Thanks to the information brought by the weak annotations, we here just

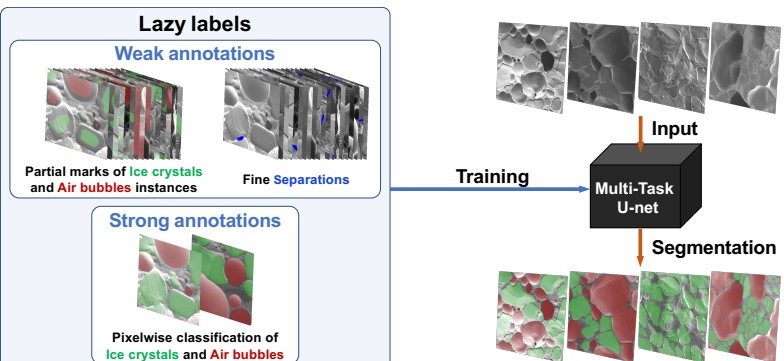

Figure 1: Multi-task learning for image segmentation with lazy labels. The figure uses Scanning Electron Microscopy (SEM) images of food microstructures as an example and demonstrates a segmentation problem of three classes, namely air bubbles (green), ice crystals (red) and background respectively. Most of the training data are weak annotations containing (i) partial marks of ice crystals and/or air bubbles instances and (ii) fine separation marks of boundaries shared by different instances. Only a few strongly annotated images are used. On the top right several SEM images are displayed. Their corresponding output, obtained with the trained network, are shown at the bottom right.

need a very small set of accurate segmentation masks, cf. bottom left part of Figure 1. To that end, we propose to refine some of the coarse labels resulting from task 1 using a semi-automatic segmentation method which requires additional manual intervention.

The three tasks are handled by a single deep neural network and are jointly optimized. The network architecture is inspired by the widely used segmentation network named U-net (Ronneberger et al., 2015). With all three tasks sharing the same contracting path, we introduce a new multi-task block for the expansive path. The network has three outputs and is fed with a combination of WL and SL described above. Since weakly and strongly annotated training data is shared between the tasks, part of the annotations are missing, especially for task 3. To accommodate for this we introduce a weighted loss function over the samples. Accurate segmentation labels for training are usually not easy to obtain, however, with our approach we demonstrate that exact labels for the whole training set are not needed for good segmentation learning performance.

We evaluate the performance of the proposed approach on two applications, namely for the segmentation of SEM images of food microstructure and stained histology images of glandular tissues.

In summary, our contributions are as follows. (1). We propose a decomposition of the segmentation problems into three tasks and a corresponding user friendly labeling strategy. (2). We develop a multi-task learning framework that learns directly from the manual labels and is trained end-to-end. Our approach outperforms state-of-the-art weakly supervised methods such as Khoreva et al. (2017) (3). The network predicts segmentation mask as well as the object inner regions and touching object interfaces. Touching objects are effectively disconnected as a side product.

## 2 RELATED WORK

In image segmentation problems, one needs to classify an image at pixel level. It is a vast topic with a diversity of algorithms being developed, including traditional unsupervised methods like $k$-means that splits the image into homogeneous regions according to image low level features, curve evolution based methods like snakes (Caselles et al., 1997), graph-cut based methods like Grabcut (Rother et al., 2004), just to name a few. Interactive approaches such as snakes or Grabcut enable getting involved users' knowledge by means of initializing regions or putting constraints on the segmentation results.

Deep convolutional neural network (DCNN) approaches have been developed for segmenting complex images, especially in the semantic setting. In particular, fully convolutional networks (FCN) (Long et al., 2015) replace the last few fully connected layers of a conventional classification network by up-sampling layers and convolutional layers, to preserve spatial information. FCNs have many variants for semantic segmentation. The DeepLab (Chen et al., 2018) uses a technique called

atrous convolution to handle spatial information together with a fully connected conditional random field (CRF) (Chen et al., 2014) for refining the segmentation results. Fully connected CRF can be used as post-processing or can be integrated into the network architecture, allowing for end-to-end training (Zheng et al., 2015).

Other types of FCNs are deep encoder-decoder networks (Badrinarayanan et al., 2017; Ronneberger et al., 2015). They have multiple up-sampling layers for better localizing boundary details. One of the most well-known models is the U-net (Ronneberger et al., 2015). It is a fully convolutional network made of a contracting path, which brings the input images into very low resolution features with a sequence of down-sampling, and an expansive path that has an equal amount of up-sampling layers. The higher resolution feature maps on the contracting path are merged with up-sampled layers via long skip connections to recover boundary information after down-sampling.

**Weakly/semi-supervised segmentation learning.** Motivated by the heavy cost of pixel-level annotation needed for fully supervised learning, there has been a growing interest in weakly supervised learning with coarse annotations for semantic segmentation in the last years. Common weak annotations include image-level labels (Huang et al., 2018; Papandreou et al., 2015; Lee et al., 2019), bounding boxes(Khoreva et al., 2017), scribbles (Lin et al., 2016) and points (Bearman et al., 2016). Most weakly supervised deep learning methods for segmentation are built on top of a classification network. The training of such networks may be realized using segmentation masks explicitly generated from weak annotations (Wei et al., 2017; Khoreva et al., 2017; Lee et al., 2019; Wei et al., 2018; Tsutsui et al., 2018). The segmentation masks can be improved recursively, which involves several rounds of training of the segmentation network (Wei et al., 2017; Jing et al., 2018; Ezhov et al., 2018).

Other techniques do not use any explicit pseudo segmentation masks as training data, but instead compute a composite loss from other guiding principles. For instance, the SEC method (Kolesnikov & Lampert, 2016) combines localization cues (Seed), classification coherence (Expand) and Constrain-to-boundary with a fully-connected CRF. Special modules could be designed to further exploit the information of sources, such as the Deep Seeded Region Growing (Huang et al., 2018) and saliency seeded region growing (Sun & Li, 2019).

Semi-supervised deep learning provides an alternative way to reduce the pixel-level annotation burden (Baur et al., 2017; Perone & Cohen-Adad, 2018). These methods require only a small amount of strongly annotated data and a large set of unlabelled data. Several training methods that combine weak annotations with a limited set of strong annotations have already been explored (Papandreou et al., 2015; Wei et al., 2018; Lee et al., 2019). The pixel-wise labels can be integrated through an additional loss function, and used along with their semi-supervised counterparts. Better weak supervision performance can be obtained by feeding the network with learned object localization maps (Wei et al., 2018; Lee et al., 2019). However, the networks are trained with image level labels from large scale datasets like ImageNet (Deng et al., 2009) or PASCAL VOC 2012 (Everingham et al., 2015).

The weakly and semi-supervised deep learning methods are also explored for object localization or segmentation on relatively smaller datasets, especially in medical imaging. For these methods, the weak annotations can be of image level (Mlynarski et al., 2018; Playout et al., 2019; Zhou et al., 2019; Shin et al., 2019) or in forms of bounding boxes (Shah et al., 2018).

**Multi-task learning.** Multi-task learning algorithms consider several related tasks to improve the overall performance, taking benefits from the underlying common information that may be ignored by a single task learning. In deep neural based multi-task models, the common information is conveyed by soft or hard parameter sharing (Ruder, 2017). Various multi-task deep learning methods have been developed for segmentation, for example, the stacked U-net for extracting roads from satellite imagery (Sun et al., 2018), the two stage 3D U-net framework for 3D CT or MR data segmentation (Wang et al., 2018a), encoder decoder networks for depth regression, semantic and instance segmentation (Kendall et al., 2018), and for building footprint segmentation (Bischke et al., 2017). Compared to these works, our method handles the tasks with a multi-task block at different feature resolutions and is designed upon both strong and weak notations.

Our work is more closely related to the multi-task learning methods in (Playout et al., 2018; 2019) for retinal lesions segmentation and (Mlynarski et al., 2018) for brain tumour segmentation. Three tasks are considered in (Playout et al., 2019): red lesion segmentation, bright lesions segmentation

and image level lesion detection. The proposed encoder-decoder architecture has one branch in the encoder part and two branches into decoder part specified for the red/bright lesion segmentation and the image level classification respectively. The network is weakly supervised as only image level labels are used in one of the training phases. In (Mlynarski et al., 2018), a U-net architecture is used for jointly segmenting and classifying brain tumours. An additional branch takes the last but one layer of a standard segmentation U-net as input, followed by a mean pooling layer, and outputs a score for the classification task.

Our network structure, similar to these architectures, shares an encoder part for all tasks. The learning is also supervised by a mixture of strong annotations and weak annotations. However, the proposed method does not include supervision from image level annotations and is more specialized in distinguishing the different object instances and clarifying their boundaries in every single image.

## 3 MULTI-TASK LEARNING FRAMEWORK

Fully supervised learning for segmentation approximates the conditional probability distribution of the segmentation mask given the image. Let $s^{(3)}$ be the segmentation mask and $I$ be the image, then the segmentation task aims to estimate $p\left(s^{(3)} \mid I\right)$ based on a set of sample images $\mathcal{I} = \{I_1, I_2, \cdots, I_n\}$ and the corresponding labels $\{s_1^{(3)}, s_2^{(3)}, \cdots, s_n^{(3)}\}$. The set $\mathcal{I}$ is randomly drawn from an unknown distribution. In our setting, having the whole set of segmentation labels $\{s_i^{(3)}\}_{1,\cdots,n}$ is impractical, and we introduce two auxiliary tasks for which the labels can be more easily generated to achieve an overall small cost on labeling.

For a given image $I \in \mathcal{I}$, we denote as $s^{(1)}$ the rough instance detection mask, $s^{(2)}$ a map containing some interfaces shared by touching objects. To create $s^{(1)}$, the contours of the objects are not treated carefully, resulting in a coarse mask that misses most of the boundary pixels, cf, the left of Figure 1. Let $\mathcal{I}_k \subset \mathcal{I}$ denote the subset of images labelled for task $k$ ($k = 1, 2, 3$). As we collect a different amount of annotations for each task, the number of annotated images $|\mathcal{I}_k|$ may not be the same for different $k$. Typically the number of images with strong annotations $|\mathcal{I}_3| \ll n$.

The set of samples in $\mathcal{I}_3$ for segmentation being small, the computation of an accurate approximation of the true probability distribution $p\left(s^{(3)} \big| I\right)$ is a challenging issue. Given that much more samples of $s^{(1)}$ and $s^{(2)}$ are observed, it is relatively easier to learn the statistics of the weak labels. Therefore, in a multi-task learning setting, one aims at approximating also the conditional probability $p\left(s^{(1)} \big| I\right)$ and $p\left(s^{(2)} \big| I\right)$ for the other two tasks, or the joint probability $p\left(\left(s^{(1)}, s^{(2)}, s^{(3)}\right) \mid I\right)$. The three tasks are connected to each other. By the definition of the detection task, it is not difficult to see that $p\left(s^{(3)} = z \big| s^{(1)} = x\right) = 0$ for $x$ and $z$ satisfying $x_{i,c} = 1$ and $z_{i,c} = 0$ for some pixel $i$ and class $c$ other than the background. The map of interfaces $s^{(2)}$ indicates small gaps between two connected instances, and is therefore a subset of boundary pixels of the mask $s^{(3)}$.

Let us consider the probabilities given by the models $p\left(s^{(k)} \big| I; \theta\right)$ ($k = 1, 2, 3$) parametrized by $\theta$ which is determined such that the models match the desired probability distributions. The parameter $\theta$ is shared among all the tasks. We do not optimize $\theta$ for each individual task, but instead consider a joint probability $p\left(\left(s^{(1)}, s^{(2)}, s^{(3)}\right) \mid I; \theta\right)$. Assuming that $s^{(1)}$ (rough under-segmented instance detection) and $s^{(2)}$ (subset of shared boundaries) are conditionally independent given image $I$, and if the samples are i.i.d, we define the maximum likelihood (ML) estimator for $\theta$ as

$$\theta_{\mathrm{ML}} = \arg\max_{\theta} \sum_{I \in \mathcal{I}} \left( \log p\left(s^{(3)} \mid s^{(1)}, s^{(2)}, I; \theta\right) + \sum_{k=1}^{2} \log p\left(s^{(k)} \mid I; \theta\right) \right). \tag{1}$$

The set $\mathcal{I}_3$ may not be evenly distributed across $\mathcal{I}$, but we assume that it is generated by a fixed distribution as well. Provided that the term $\left\{ p\left(s^{(3)} \mid s^{(1)}, s^{(2)}, I\right) \right\}_{I \in \mathcal{I}}$ can be approximated correctly by $p\left(s^{(3)} \mid s^{(1)}, s^{(2)}, I; \theta\right)$ even if $\theta$ is computed without $s^{(3)}$ specified for $\mathcal{I} \backslash \mathcal{I}_3$, then

$$\sum_{I \in \mathcal{I}} \log p\left(s^{(3)} \mid s^{(1)}, s^{(2)}, I; \theta\right) \propto \sum_{I \in \mathcal{I}_3} \log p\left(s^{(3)} \mid s^{(1)}, s^{(2)}, I; \theta\right). \tag{2}$$

If furthermore, the segmentation mask does not depend on $s^{(1)}$ or $s^{(2)}$ given $I \in \mathcal{I}_3$, and if $|\mathcal{I}_1|, |\mathcal{I}_2|$ are large enough, then from Equations (1), and (2), we approximate the ML estimator by

$$\hat{\boldsymbol{\theta}} = \arg\max_{\boldsymbol{\theta}} \sum_{k=1}^{3} \left( \alpha_k \sum_{I \in \mathcal{I}_k} \log p\left(\boldsymbol{s}^{(k)} \mid I; \boldsymbol{\theta}\right) \right) \tag{3}$$

in which $\alpha_1, \alpha_2, \alpha_3$ are non negative constants.

## 3.1 Loss function

Let the output of the approximation models be denoted respectively by $h_{\boldsymbol{\theta}}^{(1)}(I)$, $h_{\boldsymbol{\theta}}^{(2)}(I)$, and $h_{\boldsymbol{\theta}}^{(3)}(I)$, with $\left[ h_{\boldsymbol{\theta}}^{(k)}(I) \right]_{i,c}$ the estimated probability of pixel $i$ being in class $c$ of task $k$. For the label $\boldsymbol{s}^{(k)}$ of $I$, the log likelihood function for each task is decomposed into

$$\log p\left(\boldsymbol{s}^{(k)} \mid I; \boldsymbol{\theta}\right) = \sum_{i} \sum_{c \in C_k} s_{i,c}^{(k)} \log \left[ h_{\boldsymbol{\theta}}^{(k)}(I) \right]_{i,c}, \quad k = 1, 2, 3, \tag{4}$$

in which $s_{i,c}^{(k)}$ denotes the element of the label $\boldsymbol{s}^{(k)}$ at pixel $i$ for class $c$ and $C_k$ is the set of classes for task $k$. For example, for ice cream images, we have three classes including air bubbles, ice crystals and the rest (background or parts of the objects ignored by the weak labels), so $C_1, C_3 = \{1, 2, 3\}$. For the separation task, there are only two classes for pixels (belonging or not to a touching interface): $C_2 = \{1, 2\}$. According to Equation (3), the network is trained by minimizing the weighted cross entropy loss:

$$L(\boldsymbol{\theta}) = -\sum_{I \in \mathcal{I}} \sum_{k=1}^{3} \alpha_k \mathbb{1}_{\mathcal{I}_k}(I) \log p\left(\boldsymbol{s}^{(k)} \mid I; \boldsymbol{\theta}\right), \tag{5}$$

Here $\mathbb{1}_{\mathcal{I}_k}(\cdot)$ is an indicator function which is 1 if $I \in \mathcal{I}_k$ and 0 otherwise.

## 3.2 Multi-task Network

We follow a convolutional encoder-decoder network structure for the multi-task learning. The network architecture is illustrated in Figure 2. As an extension of the U-net structure for multiple tasks, we only have one contracting path that encodes shared features representation for all the tasks. On the expansive branch, we introduce a multi-task block at each resolution to support different learning purposes (blue blocks in Figure 2). Every multi-task block runs three paths, with three inputs and three corresponding outputs, and it consists of several sub-blocks.

In each multi-task block, the detection task (task 1) and the segmentation task (task 3) have a common path similar to the decoder part of the standard U-net. They share the same weights and use the same concatenation with feature maps from contracting path via the skip connections. However, we insert an additional residual sub-block for the segmentation task. The residual sub-block provides extra network parameters to learn information not known from the detection task, *e.g.* object boundary localization. The path for the separation task (task 2) is built on the top of detection/segmentation ones. It is also a U-net decoder block structure, but the long skip connections start from the sub-blocks of the detection/segmentation paths instead of the contracting path. The connections extract higher resolution features from the segmentation task and use them in the separation task.

To formulate the multi-task blocks, let $\boldsymbol{x}_l$ and $\boldsymbol{z}_l$ denote respectively the output of the detection path and segmentation path at the multi-task block $l$, and let $\boldsymbol{c}_l$ be the feature maps received from the contracting path with the skip connections. Then for task 1 and task 3 we have

$$\boldsymbol{x}_{l+1} = F_{W_l}(\boldsymbol{x}_l, \boldsymbol{c}_l), \qquad \boldsymbol{z}_{l+\frac{1}{2}} = F_{W_l}(\boldsymbol{z}_l, \boldsymbol{c}_l), \qquad \boldsymbol{z}_{l+1} = \boldsymbol{z}_{l+\frac{1}{2}} + F_{W_{l+\frac{1}{2}}}(\boldsymbol{z}_{l+\frac{1}{2}}), \tag{6}$$

in which $W_l, W_{l+1/2} \in \boldsymbol{\theta}$ are parameters of the network and $F_{W_l}$, $F_{W_{l+\frac{1}{2}}}$ are determined respectively by a sequence of layers of the network (Cf. the small grey blocks on the right of Figure 2). For task 2 the output at $l^{\text{th}}$ block $\boldsymbol{y}_{l+1}$ is computed by $\boldsymbol{y}_{l+1} = G_{\tilde{W}_l}(\boldsymbol{z}_{l+1}, \boldsymbol{y}_l)$ with additional network parameters $\tilde{W}_l \in \boldsymbol{\theta}$. Finally, after the last multi-task block, softmax layers are added, outputting a probability map for each task.

**Implementation details.** We implement a multi-task U-net with 6 levels of spatial resolution and input images of size $256 \times 256$. A sequence of down-sampling via max-pooling with pooling

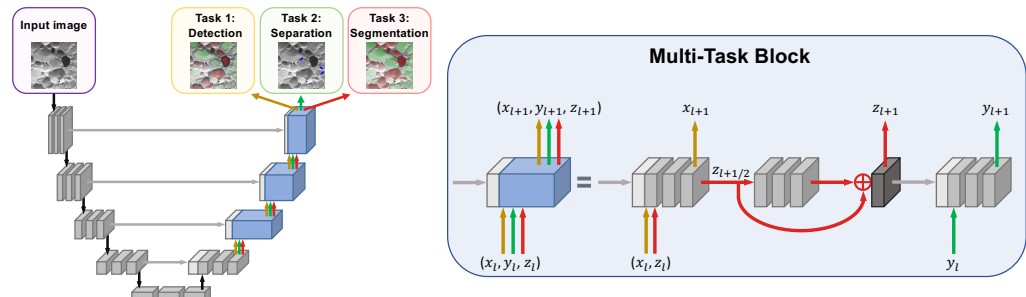

Figure 2: Architecture of the multi-task U-net. The left part of the network is a contracting path similar to the standard U-net. For multi-task learning, we construct several expansive paths with specific multi-task blocks. At each resolution, task 1 (Detection in yellow) and task 3 (Segmentation in red) run through a common sub-block, but the red path learns an additional residual to better localize object boundaries. Long skip connections with the layers from contracting path are built for yellow/red paths via concatenation. Task 2 (Separation, in green) mainly follows a separated expansive path, with its own up-sampled blocks. A link with the last layer of task 3 is added via a skip connection in order to integrate accurate boundaries in the separation task.

size $2 \times 2$ is used for the contracting path of the network. Different from the conventional U-net (Ronneberger et al., 2015), each small gray block (see Figure 2) consists of a convolution layer and a batch normalization (Ioffe & Szegedy, 2015), followed by a leaky ReLU activation with a leakiness parameter $0.01$. The same setting is also applied to gray sub-blocks of the $4$ multi-task blocks. On the expansive path of the network, feature maps are up-sampled (with factor $2 \times 2$ ) by bilinear interpolation from a low resolution multi-task block to the next one.

### 3.3 METHODS FOR LAZY LABELS GENERATION

We now explain our strategy for generating all the lazy annotations that are used for training the multi-task U-net. We introduce our method with a data set of ice cream SEM images but any other similar dataset could be used. Typical images of ice cream samples are shown in the top row of the left part of Figure 3. The segmentation problem is challenging since the images contain densely distributed small object instances (*i.e.*, air bubble and ice crystals), and poor contrast between the foreground and the background. The sizes of the objects can vary significantly in a single sample. Textures on the surfaces of objects also appear.

As a first step, scribble-based labelling is applied to obtain detection regions of air bubbles and ice crystals for task 1. This can be done in a very fast way as no effort is put on the exact object boundaries. We adopt a lazy strategy by picking out an inner region for each object in the images (see *e.g.*, the second row of the left part of Figure 3). Though one could get these rough regions as accurate as possible, we delay such refinement to task 3, for better efficiency of the global annotation process. Compared to the commonly used bounding box annotations in computer vision tasks, these labels give more confidence for a particular part of the region of interest.

In the second step, we focus on tailored labels for those instances that are close one to each other (task 2), without a clear boundary separating them. Again, we use scribbles to mark their interface. Examples for such annotations are given in Figure 3 (top line, right part) The work can be carried out efficiently especially when the target scribbles have a sparse distribution. On the other hand, as no labelling is needed for the objects that are well separated, we can collect sufficient labelled images in a limited amount of time and cover the complex ice cream sample conditions. Lazy manual labeling of tasks 1 and 2 are done independently. It follows the assumption made in Section 3 that $s^{(1)}$ and $s^{(2)}$ are conditionally independent given image $I$.

The precise labels for task 3 are created using interactive segmentation tools. Starting from the rough (inner) regions of task 1, a natural idea is to let these regions grow and stop when the boundaries are reached. This can be done with geodesic active contours (Caselles et al., 1997). Unfortunately, such a method fails to capture sharp corners and the contour evolution tends to ignore boundaries with low

contrast. The annotation then requires frequent and time consuming user interaction. Instead, we use Grabcut (Rother et al., 2004; Hong et al., 2015) a graph-cut based method. The initial labels obtained from the first step give a good guess of the whole object regions. The Grabcut works well on isolated objects. However, it gives poor results when the objects are close to each other and have boundaries with inhomogeneous colors. As corrections may be needed for each image, only a few images of the whole dataset are processed. A fully segmented example is shown in the last row of Figure 3.

## 4 EXPERIMENTS

In this section, we demonstrate the performance of our approach using two different datasets. For both datasets we use strong labels (SL) as well as weak labels (WL). We prepare the labels and design the network as described in Section 3. The overall method is summarized with the 2 procedures presented in Algorithm 1 (see the appendix).

### 4.1 SEM IMAGES OF ICE CREAM

Scanning Electron Microscopy (SEM) constitutes the state-of-the-art for analysing food microstructures as it enables the efficient acquisition of high quality images for food materials, resulting into a huge amount of image data available for analysis. However, to better delineate the microstructures and provide exact statistical information, the partition of the images into different food components and instances is needed. The structures of food, especially soft solid materials, are usually complex which makes automated segmentation a difficult task. Some SEM images of ice cream in our dataset are shown on the top right of Figure 1. A typical ice cream sample consists of air bubbles, ice crystals and a concentrated unfrozen solution. In most situations, the air bubbles and ice crystals appear as foam in the images, while the solution fills the gaps between them. We treat the solution as the background and aim at detecting and computing a pixel-wise classification for each air bubbles and ice crystals instances.

The set of ice-cream SEM dataset consists of 38 images that are split into three sets (53% for training, 16% for validation and 31% testing respectievly). Each image contains a rich set of instances with an overall number of instances around 13300 for 2 classes (ice crystals and air bubbles). For comparison, the PASCAL VOC 2012 dataset has 27450 objects in total for 20 classes.

For training the network, data augmentation is applied to prevent over-fitting. The size of the raw images is $960 \times 1280$. They are rescaled and rotated randomly, and then cropped into an input size of $256 \times 256$ for feeding the network. Random flipping is also performed during training. The network is trained using Adam optimizer (Kingma & Ba, 2014) with a learning rate $r = 2 \times 10^{-4}$ and a batch size of 16.

In the inference phase, the network outputs for each patch a probability map of size $256 \times 256$. The patches are then aggregated to obtain a probability map for the whole image. In general, the pixels near the boundaries of each patch are harder to classify. We thus weight the spatial influence of the patches with a Gaussian kernel to emphasize the network prediction at patch center.

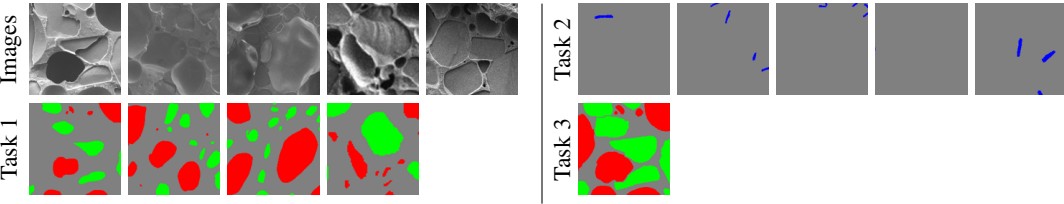

Figure 3: Example of annotated images. Some of the annotations are missing because not all images are labelled for task 1 and task 2. The marks in red are for air bubbles and the ones in green are for ice crystal instances. The blue curves on the third row are labels for interfaces of touching objects.

We now evaluate the multi-task U-net and compare it to the traditional single task U-net. The performance of each model is tested on 12 images, and average results are shown in Table 1. In the table, the dice score for a class $c$ is defined as $d_c = 2 \sum_i x_{i,c} y_{i,c} / (\sum_i x_{i,c} + \sum_i y_{i,c})$ where $\boldsymbol{x}$ is the computed segmentation mask and $\boldsymbol{y}$ the ground truth.

| The models | air bubbles | ice crystals | Overall |
|---|---|---|---|
| U-net on WL | 0.725 | 0.706 | 0.716 |
| U-net on SL | 0.837 | 0.794 | 0.818 |
| PL approach | 0.938 | 0.909 | 0.924 |
| Multi-task U-net | **0.953** | **0.931** | **0.944** |

Table 1: Dice scores of segmentation results on the test images of SEM images of ice cream dataset.

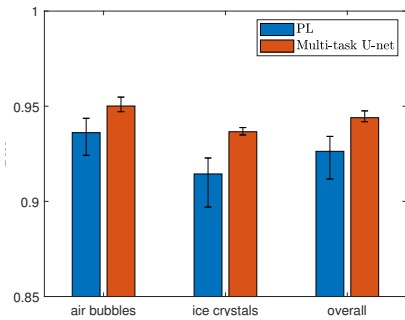

Figure 4: The error bars for the PL and multi-task U-net, each computed from 8 different experiments

We train a single task U-net (*i.e.*, without the multi-task block) on the weakly labelled set (task 1), with the 15 annotated images. The single task U-net on weak annotations gives an overall dice score at 0.72, the lowest one among the three other methods tested. One reason for the low accuracy of the single task U-net on weak (inaccurate) annotations is that in the training labels, the object boundaries are mostly ignored. Hence the U-net is not trained to recover them, leaving large parts of the object not recognized. Second, we consider strong annotations as training data, without the data of the other tasks, *i.e.* only 2 images with accurate segmentation masks are used. The score of the U-net trained on SL is only 0.82, which is significantly lower than the 0.94 obtained by the multi-task U-net.

We also compare our multi-task U-net results with one of the major weakly supervised approaches that makes use of pseudo labels (PL) (see e.g., Khoreva et al. (2017); Jing et al. (2018)). In these approaches, the pseudo segmentation masks are created from WLs and are used to feed a segmentation network. Following the work of Khoreva et al. (2017), we use the Grabcut method to create the PLs, and in our setting the partial masks are used to initialize the Grabcut. For the small subset of images that are strongly annotated, the full segmentation masks are used instead of PLs. The PL are created without human correction, and then used for feeding the segmentation network. Here we use the baseline single task U-net as the segmentation network in order to make comparison with the multi-task U-net.

Our multi-task U-net outperforms the PL approach as shown in Table 1. Figure 4 displays the error bars for the two methods with dice score collected from 8 different runs. A significant limitation of the PL method here is that its performance relies on the tools used for pseudo segmentation mask generations. Having common errors, instead of random ones, on the object boundaries in the training data, the segmentation network of PL also learns to have those patterns in the prediction. The images in the left part of Figure 5 show that the predicted label of an object tends to merge with some background pixels when there are edges of another object nearby. Similar errors from the GrabCut are illustrated in the appendix.

Besides the number of pixels that are correctly classified, the separation of touching instances is also of interest. In additional to the dice scores in Table 1, we study the learning performance of multi-task U-net on task 2, which specializes in the separation aspect. The test results on the 12 images give an overall precision of 0.70 of the detected interfaces, while 0.82 of the touching objects are recognized. We show some examples of computed separations and ground truth in the right part of Figure 5. For the detection task, the network predicts a probability map for the inner regions of the object instances. An output of the network is shown in Figure 6. With partial masks as coarse labels for this task, the network learns to identify the object instance as shown in the figure.

We finally consider the work of Bearman et al. (2016) that investigate the cost related to different types of annotations. Based on the data reported (Bearman et al., 2016) and our estimated annotation time, the collecting of WL for detection is considerably (more than 6x) faster than obtaining SL. For a fair comparison with the baseline U-net we use a larger ratio of SL for the single task learning accordingly (since no WL is used here), and the results are reported in Table 2. The proposed method still outperforms the U-net by a large margin on similar annotation time budgets, and additional SL after the first 10% do not help significantly.

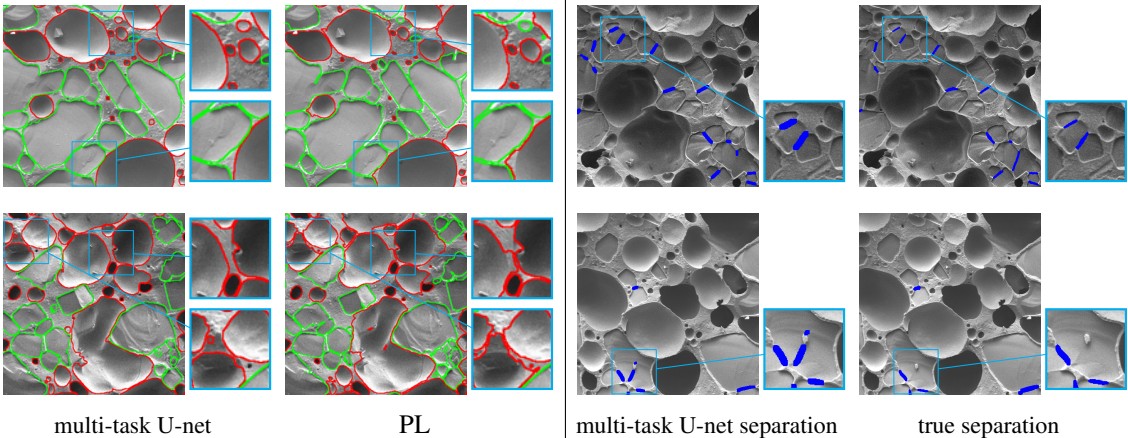

|  |  |  |  |
|---|---|---|---|
| multi-task U-net | PL | multi-task U-net separation | true separation |

Figure 5: Segmentation and separation results (best viewed in color). Left: the computed contours are shown in red for air bubbles and green for ice crystals. While multi-task U-net and PL supervised network both have good performance, PL misclassifies the background near object boundaries. Right: Examples of separation by the multi-task U-net and the ground truth.

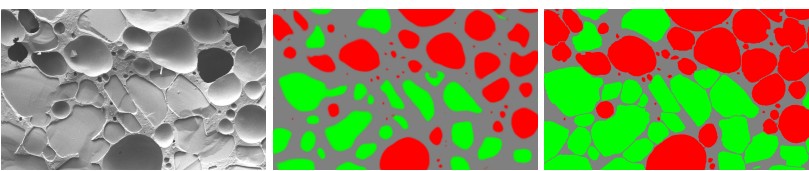

Figure 6: The image (left), the predicted probability map (middle) from the detection task, and the ground truth segmentation mask (right). The red and green on the middle and right images stand for air bubble and ice crystals respectively.)

## 4.2 H&E-STAINED IMAGE DATESET FOR GLAND SEGMENTATION

We also apply the approach to the segmentation of tissue in histology images. In this experiment, we use the GlaS challenge dataset (Sirinukunwattana et al., 2017) that consists of 165 H&E stained images. The dataset is split into three parts, with 85 images for training, and 60 for offsite test and 20 images for onsite test (we will call the latter two sets Test part A and Test part B respectively in the following).

Apart from the SL available from the dataset, we create a set of a weak labels for the detection task and separation task (as illustrated in Appendix, Figure 11). These weak labels together with a part

|  | methods | labels | dice score |
|---|---|---|---|
| Annotation budget 1 | multi-task | 10% SL + WL | **0.944** |
|  | single task | 20% SL | 0.882 |
| Annotation budget 2 | multi-task | 20% SL + WL | **0.948** |
|  | single task | 30% SL | 0.913 |
| Annotation budget 3 | multi-task | 50% SL + WL | **0.949** |
|  | single task | 75% SL | 0.940 |

Table 2: We compare the two methods under similar annotation time budgets. In each budget, two different combinations of SL and WL that take similar annotation time are used. The WL in this table contains 75% labels for the detection task and 100% labels for the separation task. From budget 1 to budget 3, we increase the amount of labels (that means more annotation time is needed) in the training data, and the dice score is reported for each case.

| SL Ratio | | 2.4% | 4.7% | 9.4% | 100% |
|---|---|---|---|---|---|
| Test Part A | Ours | **0.866** | **0.889** | **0.915** | **0.921** |
| | U-net | 0.700 | 0.749 | 0.840 | 0.915 |
| | MDUnet | | | | 0.920 |
| Test Part B | Ours | **0.751** | **0.872** | **0.904** | **0.910** |
| | U-net | 0.658 | 0.766 | 0.824 | 0.898 |
| | MDUnet | | | | 0.871 |

Table 3: Average dice score for segmentation of gland. Our method uses both SL and WL. The ratio of strong labels (SL) is increased from 2.4% to 100%, and the scores of the methods are reported here for two parts A and B of the test sets, as split in Sirinukunwattana et al. (2017).

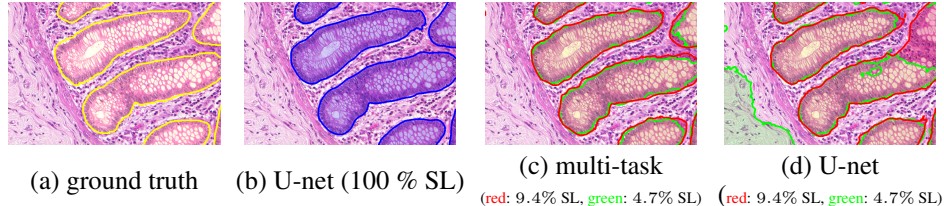

(a) ground truth     (b) U-net (100 % SL)     (c) multi-task     (d) U-net

(red: 9.4% SL, green: 4.7% SL)    (red: 9.4% SL, green: 4.7% SL)

Figure 7: Segmentation results on the gland dataset (best view in color). The ground truth and the results. For (c) and (d), Red contour denotes the results from 9.4% strong labels; Green contour denotes results from 4.7% strong labels

of the strong labels are used for training the multi-task U-net as illustrated in Algorithm 1 (in the appendix).

In this experiment, we test the algorithm on different ratios of SL, and compare it with the baseline U-net, and the Multi-scale Densely Connected U-Net (DMUnet) (Zhang et al., 2018). The results on two sets of test data are reported in Table 3. As the SL ratios increase from 2.4% to 9.4%, an improvement of performance of the multi-task U-net is gained, and when it reaches 9.4% SL, the multi-task framework achieves comparable score with the fully supervised version. We emphasize that the 9.4% SL and WL are at a much lower annotation cost than that of the 100% SL used for fully supervised learning. Example of segmentation results are displayed in Figure 7.

## 5    CONCLUSION

In this paper, we develop a multi-task learning framework for image segmentation problems, which relaxes the requirement for numerous and accurate annotations to train the network. It is suitable for segmentation problem with a dense population of object instances. The model separates the segmentation problem into three smaller tasks. One of them is dedicated to the instance detection and therefore do not need exact boundary information. This gives potential flexibility as one could concentrate on the classification and rough location of the instances during data collection. The second one focuses on the separation of objects sharing a common boundary. The final task aims at extracting pixel-wise boundary information. Thanks to the information shared within the multi-task learning, this accurate segmentation can be obtained using very few annotated data.

Our model learns directly the statistics of WL as auxiliary tasks, so no further processing steps are needed before training the network. For the partial masks that ignore boundary pixels, the annotation can also be done when the boundaries of object are hard to detect. As a small amount of SL is needed and the collection of WL can be fast and cheap, the proposed framework is potentially effective for applications with growing datasets. The weakly annotated set for detection purpose could be augmented if necessary and the new images could easily be incorporated into our end-to-end framework.

ACKNOWLEDGMENTS

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

# A  APPENDIX

## A.1  THE ALGORITHM

We summarize the overall procedures, including the steps of creating the labels and the multi-task training into Algorithm .

---

**Algorithm 1** Multi-task learning for segmentation with lazy labels

---

1: **procedure** LAZY LABELS($\mathcal{I}$)        $\triangleright$ Choose $\mathcal{I}_1, \mathcal{I}_2, \mathcal{I}_3 \subset \mathcal{I}$, $|\mathcal{I}_3|$ can be relatively small.
    **Input:** set of images $\mathcal{I}$
2:     Select inner regions for each object in $\mathcal{I}_1$ for the detection task
3:     Indicate scribbles on images of $\mathcal{I}_2$ for the separation task
4:     Generate a few pixel-wise labels $\mathcal{I}_3$ from $\mathcal{I}_1$ using interactive segmentation tools (e.g., Grabcut)
5:     **return** $\mathcal{I}_1, \mathcal{I}_2, \mathcal{I}_3$ and the labels



1: **procedure** MULTI-TASK U-NET TRAINING($\mathcal{I}_k, \boldsymbol{s}_i^{(k)}, \alpha_k$,r)      $\triangleright$ $\boldsymbol{s}_i^{(k)}$ denote the labels
    **Input:** labelled sets $\mathcal{I}_k$, loss function weights $\alpha_k$ for $k = 1, 2, 3$, Adam parameters $r$, mini-batch size $m$.
2:     Set the $1^{\text{st}}$ and $2^{\text{nd}}$ momentum vectors $\boldsymbol{m}, \boldsymbol{v}$ as zeros.
3:     Initialize the multi-task U-net parameter $\boldsymbol{\theta}$.
4:     **while** Termination criterion is not satisfied **do**
5:         Obtain a mini-batch $(I_1, \boldsymbol{s}_1^{(k)}), \cdots, (I_m, \boldsymbol{s}_m^{(k)})$.    $\triangleright$ $\boldsymbol{s}_i^{(k)}$ can be unknown for some $(i, k)$.
6:         Compute the gradient   $\boldsymbol{g} \leftarrow \nabla_{\boldsymbol{\theta}} \frac{1}{m} \sum_{k=1}^{3} \sum_{i=1}^{m} \alpha_k \mathbb{1}_{\mathcal{I}_k}(I_i) \log p\left(\boldsymbol{s}_i^{(k)} \mid I_i; \boldsymbol{\theta}\right)$
7:         $(\boldsymbol{\theta}, \boldsymbol{m}, \boldsymbol{v}) \leftarrow \text{Adam}_r(\boldsymbol{\theta}, \boldsymbol{m}, \boldsymbol{v}, \boldsymbol{g})$      $\triangleright$ $\text{Adam}_r(\cdot)$ is an Adam iteration
8:     **return** $\boldsymbol{\theta}$

---

## A.2   FURTHER RESULTS FOR THE ICE CREAM IMAGE DATASET

Figure 8 shows the Dice scores on the 12 test images from Figure 9. One reason for the low accuracy of the single task U-net on weak (inaccurate) annotations is that the U-net is not trained to recover object boundaries. Therefore, the corresponding dice score on every test image is low, as shown by the blue curve. On the other hand, the network trained on strong annotations has a very good performance on only a few test images, as shown by the red line. Learning from both strong and weak annotations, the multi-task U-net improves from the other two, and the dice scores are between $0.9$ to $0.95$ for all of the images.

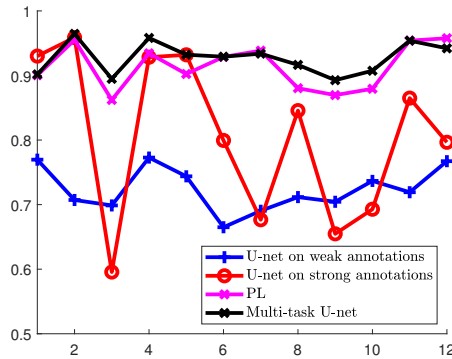

Figure 8: Overall dice scores for the air bubbles and ice crystals plotted versus the test image number. We show the scores for the U-net baseline with weak annotations, strong annotations and pseudo labels (PL) respectively, compared along with the multi-task U-net

### A.2.1   ADDITIONAL RESULTS

We compare the segmentation results given by the multi-task U-net with the ones obtained using GAC Caselles et al. (1997) and Grabcut Rother et al. (2004) (Figure 10). To apply GAC and Grabcut, we convert the problem into a single class and single object segmentation problem. We use the rough inner regions from the first step of the labelling to estimate a bounding box around each object. We then apply GAC and Grabcut inside each box. The box-wise results are aggregated to get the segmentation mask of the whole image.

Both GAC and Grabcut fail to capture the boundaries in various situations, such as intensity variations within the objects and background fragments connected to the objects. This demonstrates that a precise labelling requires many additional user interventions and thus the interest of our lazy labelling.

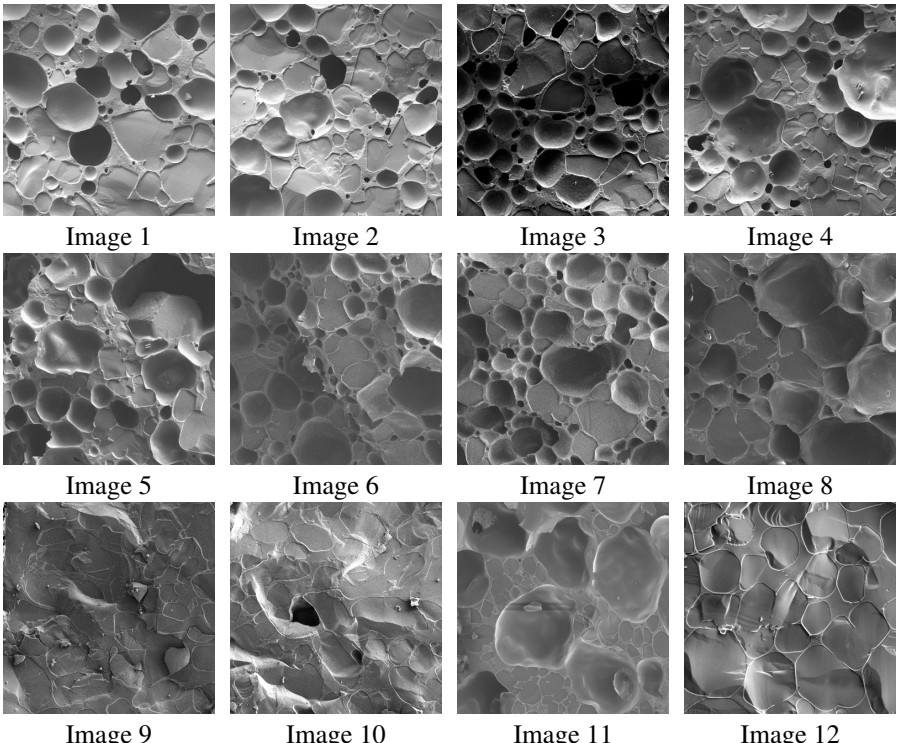

Figure 9: Twelve patches from the twelve test images respectively.

As shown in the third column of Figure 10), the Grabcut repeatedly misclassified small pieces of background as parts of the object especially when two objects are close to each other with some weak boundaries. The PL learning methods also give this kind of error in the inference phase. This means that the error has been learned by the segmentation network. The manual detection and correction of such small misclassified regions is a tedious task as well, and therefore is a limitation of the PL approach.

## A.3    THE H&E-STAINED IMAGE DATESET FOR GLAND SEGMENTATION

Figure 7 shows examples of WL and SL that are used for the segmentation of gland.

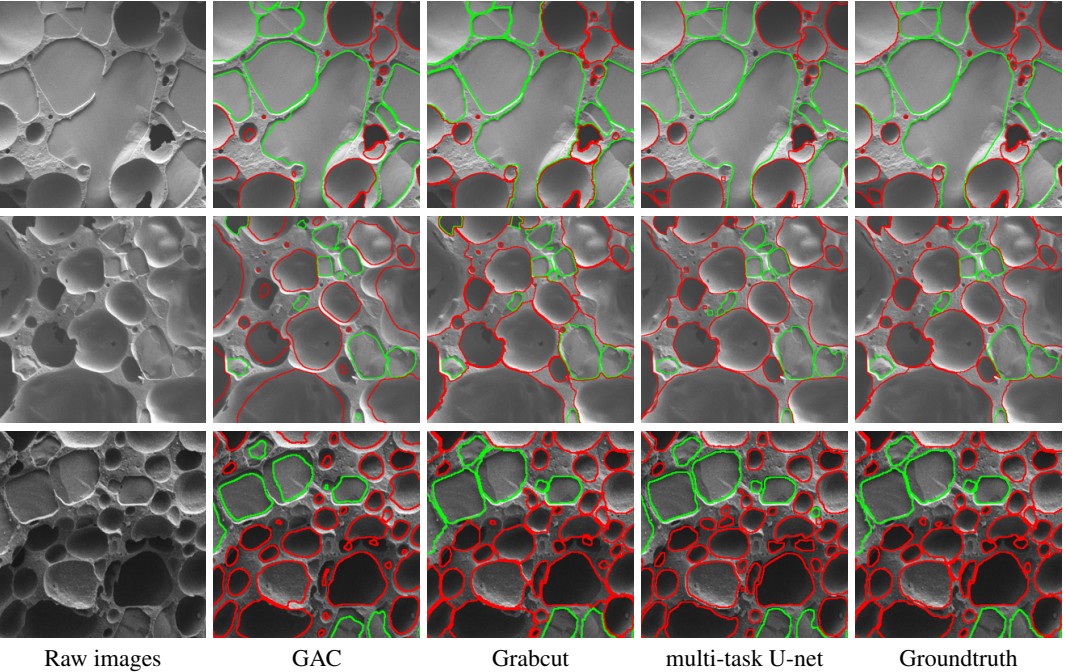

Figure 10: Comparison of multi-task U-net with the GAC and Grabcut methods using the weak labels of the validation data. Green curves: contour of the ice crystals. Red curves: air bubbles.

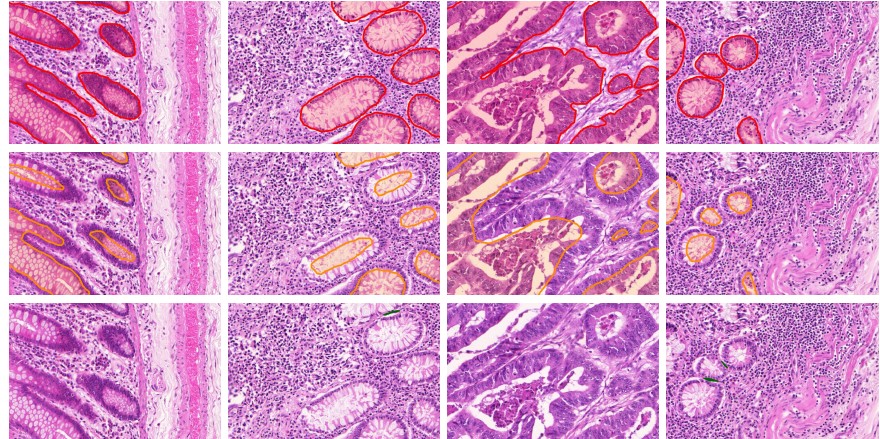

Figure 11: Gland segmentation dataset with SL in the first row (highlighted in red), WL in the second row for detection (in orange) and weak labels in the third row for separation (in dark green). The labels for separation can be sparse with some images (see the third row) having zero annotation as the instances are not touching each other.

