# OpenReview forum: "A multi-task U-net for segmentation with lazy labels"
_ICLR.cc/2020/Conference — Reject_

### Official Review · AnonReviewer3 · 2019-10-20
**Official Blind Review #3**

**Rating:** 6

**Review:**

The submission presents a neural network for multi-task learning on sets of labeled data that are largely weakly supervised (in this case, partially segmented instances), augmented by comparatively fewer fully supervised annotations.
The multiple tasks are designed to make use of both the weak as well as as full (‘strong’) labels, such that performance on fully annotated machine-generated output is improved.

As noted in the related work section (Section 2), multi-task methods aim to use benefits from underlying common information that may be ignored in a single-task setting. The network presented here is quite similar to most of these multi-task approaches: a common feature encoder, and partially distinct feature decoding and classification parts.
The (minor) novelty mainly comes from the distinct types of weak/strong annotation data fed here: instance scribbles, boundary scribbles, and (some or few) full segmentations.

The submission is overall well written and provides sufficient clarity and a good overview of the approach.
Section 3 presents a probabilistic decomposition of the proposed architecture. With some fairly standard assumptions and simplifications, the loss in Eq. 3 becomes rather straightforward (weighted cross entropy)
The actual network architecture described in Section 3.2 takes a standard U-Net as a starting point and modifies it in a fairly targeted way for the different expected types of annotations. These annotations (Section 3.3) are cheaper than full labels on a same-size dataset; it is not completely clear, however, if the mentioned scribbles need to capture each instance in the training set, or if some can also be left out. Without this being explicitly mentioned, I will assume the former.

The experimental evaluation is done reasonably well, although I am not familiar with any of the presented data sets. The SES set seems to be specific to the submission, while the H&E data set has been used at least one other relevant publication (Zhang et al.). My main issue here is that at least on the SES set, which does not seem to be that large, the score difference is not that big, so dataset bias could play some part (which is unproven, but so is the opposite).
Experimental evaluation does not leave the low-number-of-classes regime, and I’m left wondering how the method might compare on a semantically much richer data set, e.g. Cityscapes. Finally, unmodified U-Net is by now a rather venerable baseline, so I’m also wondering how the proposed multi-task learning could be used in other (more recent) architectures, i.e. whether the idea can be generalized sufficiently.

While I think the ideas per se have relatively minor novelty, the combination seems novel to me, and that might warrant publication.

**Experience Assessment:**

I have published in this field for several years.

**Review Assessment: Checking Correctness Of Derivations And Theory:**

I assessed the sensibility of the derivations and theory.

**Review Assessment: Checking Correctness Of Experiments:**

I assessed the sensibility of the experiments.

**Review Assessment: Thoroughness In Paper Reading:**

I read the paper at least twice and used my best judgement in assessing the paper.

---

> ### Author Response · Authors · 2019-11-09
> **Response to Official Blind Review #3**
>
>
> We thank the reviewer for the comments and feedback.
>
> >> it is not completely clear, however, if the mentioned scribbles need to capture each instance in the training set, or if some can also be left out.
>
> We used different subsets of images (i.e., $\mathcal{I}_1$, $\mathcal{I}_2$ and $\mathcal{I}_3$) labeled for different tasks. Therefore, the scribbles (WL) do not need to capture every instance in the training sets. However, we do have an assumption that most of the images are weakly labelled. This is motivated by the fact that the WL are collected in a much cheaper way than the SL. As we mentioned in Section 3, the missing labels (either SL or WL) are incorporated using a weighted loss function (given by Equation (5)).
>
>
> >> Experimental evaluation does not leave the low-number-of-classes regime, and I’m left wondering how the method might compare on a semantically much richer data set, e.g. Cityscapes
>
> The successful examples we have shown make the proposed model relevant for a wide range of applications with a low number of classes (for example biomedical imaging data) and namely the practical problems we are working on.
> It would indeed be interesting to try the approach for a problem with a larger number of classes, but it would require specific data. Our method has been designed to address problems in which datasets have fewer images, not highly contrasted images and where finding very accurate contours is essential. Most semantically rich and large datasets, such as Cityscapes, have better contrast and the annotations are not always accurate enough.
>
>
>
> >> Finally, unmodified U-Net is by now a rather venerable baseline, so I’m also wondering how the proposed multi-task learning could be used in other (more recent) architectures, i.e. whether the idea can be generalized sufficiently.
>
> We have not explored all possible architectures in our main segmentation network, but the idea of decomposition of the segmentation problem into the mentioned smaller tasks is to a certain extent agnostic to the underlying architecture and readily applicable to a lot of different settings. The choice of network architectures has some flexibility depending on the problem at hand, and the multi-task learning block could be used in other architectures that have multiple levels of resolution design (such as SegNet).

---

### Official Review · AnonReviewer1 · 2019-10-22
**Official Blind Review #1**

**Rating:** 1

**Review:**

This paper proposes a method for semantic segmentation using "lazy" segmentation labels. Lazy labels are defined as coarse labels of the segmented objects. The proposed method is a UNET trained in a multitask fashion whit 3 tasks: object detection, object separation, and object segmentation. The method is trained on 2 datasets: air bubbles, and ice crystals. The proposed method performs better than the same method using only the weakly supervised labels and the one that only uses the sparse labels.

The novelty of the method is very limited. It is a multitask UNET. The method is compared with one method using pseudo labels. However, this method is not SOTA. Authors should compare with current methods such as:
 - Where are the Masks: Instance Segmentation with Image-level Supervision
 - Instance Segmentation with Point Supervision
 - Object counting and instance segmentation with image-level supervision
 - Weakly supervised instance segmentation using class peak response
 - Soft proposal networks for weakly supervised object localization
 - Learning Pixel-level Semantic Affinity with Image-level Supervision for Weakly Supervised Semantic Segmentation
These methods can use much less supervision (point-level, count-level or image-level) and may work even better.

The method should be compared on standard and challenging datasets like Cityscapes, PASCAL VOC 2012, COCO, KITTI...



**Experience Assessment:**

I have published in this field for several years.

**Review Assessment: Checking Correctness Of Derivations And Theory:**

I assessed the sensibility of the derivations and theory.

**Review Assessment: Checking Correctness Of Experiments:**

I carefully checked the experiments.

**Review Assessment: Thoroughness In Paper Reading:**

I read the paper at least twice and used my best judgement in assessing the paper.

---

> ### Author Response · Authors · 2019-11-09
> **Response to Official Blind Review #1 (Part 2)**
>
>
> >> The method should be compared on standard and challenging datasets like Cityscapes, PASCAL VOC 2012, COCO, KITTI…
>
> There are various reasons why we think the current datasets are more suitable for this work.
> 1). The mentioned standard datasets are about images from natural scenes and they generally have better contrast between the objects and the background compared to the datasets that we used. Indeed, one key objective of our work is to obtain accurate object contours for images in poor contrast (for instance. images in Figure 9).
> 2). We wish to develop methods that are applicable for application domains where only relatively small training sets are available, as the ones demonstrated in the paper.
> 3).  We are focusing on problems where finding a precise segmentation is important. On the large datasets mentioned by the reviewer, the annotations are not always accurate enough:
>    - Cityscapes Dataset: The legs of the man on the motorcycle in Example Weimar 1 in https://www.cityscapes-dataset.com/examples/
>    - COCO 2019: Legs of the tennis player in http://cocodataset.org/#detection-2019
>    - In KITTI, cars are often not segmented when touching image boundaries (second line of Fig. 1 page 16 of https://arxiv.org/pdf/1604.05096.pdf, Fig 9 (e) in https://arxiv.org/pdf/1908.11656.pdf )

---

> ### Author Response · Authors · 2019-11-09
> **Response to Official Blind Review #1 (Part 1)**
>
>
> We thank the reviewer for the feedback and suggestions.
>
> >> The method is trained on 2 datasets: air bubbles, and ice crystals.
> - In this work we tested our method on two datasets, namely the SEM dataset (having three classes, air bubbles, ice crystals and the background) and H&E dataset (gland histology images).
>
> >>  Authors should compare with current methods such as Where are the Masks: Instance Segmentation with Image-level Supervision ...
>
> We thank the referee for the suggestions.
> Given the data, however,  we consider point-level, count-level or image-level supervision - as in the works listed by the referee - not well adapted due to various reasons that we will explain in the subsequent discussions.
>
> In paragraph “Weakly/semi-supervised segmentation learning” in our paper, we already included a review on papers which contain these kinds of supervision. The mentioned papers have the same limitations with respect to our approach than the approaches in the papers mentioned by the referee:
> - First notice that our weak labels can be easily converted into labels in the form of points, counting, or image-level with a certain loss of information. However, our aim is to get a good trade-off between the annotation effort and the segmentation performance.
> - The image level or count level labels do not directly encode the location of an instance or information for separating touching instances. Therefore, supervision based on these labels requires in general a much larger number of images, which is not feasible for applications with datasets of limited size.
> - Our method is applicable to image where instance counting or image-level labelling is not a trivial task. Indeed, at the border of the images, most of the objects are partially visible (cf. Figure 3), and instance counting might be less straightforward. Also it is not very easy to manually count the number of objects for datasets in the paper where a single image contains hundreds of densely distributed objects.
> - In point-level supervision, more information about the object location is available. Based on this, we move a step further by incorporating the rough regions into the label with a slightly larger cost. This information helps the network to predict coarse regions of the instances apart from their location as illustrated in Figure 5.
>
> We claim that our approach provides a good alternative to the current weak supervision strategies.  The form of WL allows the user some freedom for making the annotations, and the raw WL (with noise) can be directly used in an end-to-end training setting.
>
> Regarding the papers mentioned by the reviewer, they are interesting pieces of work. Nevertheless, we also observed some major differences between them and our paper.
> - The method in the 2nd paper does not directly tackle our main issue.  (i) The proposed network only predicts localization of pixels and embedding vectors rather than masks.  (ii) The resulting masks, however, are selected from a set of pseudo masks generated by an object proposal method (SharpMask, ECCV 2016). The segmentation results, as shown in Figure 1 and Figure 6 of the paper, do not fit the object contours well. Our work, in contrast, is dedicated to obtaining accurate contours which is not addressed by this paper.
> - The method in the 1st paper trains a segmentation network using pseudo masks generated by a classifier trained with image-level labels. We noticed that the image level supervision might not extend well to our experiments as we discussed above. Furthermore, the segmentation results (Figure 4) do not have a good match of object contours, but this is important for our problems.
> - The 3rd to the 6th papers uses image-level supervision. However, it is not clear how to adapt the classifiers such that they are efficient for datasets with a limited number of images and each image containing hundreds of objects.

---

### Official Review · AnonReviewer2 · 2019-10-28
**Official Blind Review #2**

**Rating:** 6

**Review:**

Summary:
This paper addresses the problem of learning segmentation models in the presence of weak and strong labels of different types. This is an important problem that arises while learning in data-scarce settings. The presented approach optimizes jointly over the various types of labels by treating each one as a task. They extend the U-net architecture to incorporate the different tasks.

Prior work:
There has been other work on incorporating multi-resolution or different types of labels. Here is one that can be cited:
Label super-resolution networks (https://openreview.net/forum?id=rkxwShA9Ym)

Major comments:
- The motivation for the specific structure of the multi-task blocks is not clear
- The object boundaries labels can be noisy (i.e s(2) can have noise). How does model deal with this?
- Is it the case that every image in I_3 is completely labeled - i.e all segments/classes marked?
- The assumption that s(3) is independent of s(1) and s(2) is not true. Instead of constraining the model to learn masks that respect the various types of labels, it seems they learn from each source independently. It is not clear how the sharing of parameters in the multitask block helps.
- Can they comment on the applicability of the prior work suggested above?

Minor comments:
- How do the rough labeling tools work on biomedical data where the objects are more heterogenous patterns where different labels can have very different distribution of pixels. How well will their method generalize in such settings?
- Can this work be used for segmentation and prediction on crop data?

Results:
- It seems as if the improvement over the PL baseline (pseudo labels) is incremental? Can the authors provide error bars so the reader knows what the significance of the results is?
- Can they give a more thorough comparison in terms of human effort? It is interesting to note that only 2 images give 0.82. Would 3 images give 0.94? They need to show the trade-off between additional effort vs gains in performance.
- What is the performance of MT U-net without the SL images (i.e without task-3)? Table-2 does give some intuition, but authors should add another row with multitask WL
- Table-3: How well does MDUnet do with 9.4% SL data?




**Experience Assessment:**

I have read many papers in this area.

**Review Assessment: Checking Correctness Of Derivations And Theory:**

N/A

**Review Assessment: Checking Correctness Of Experiments:**

I carefully checked the experiments.

**Review Assessment: Thoroughness In Paper Reading:**

I read the paper thoroughly.

---

> ### Author Response · Authors · 2019-11-09
> **Response to Review #2 (Part 2)**
>
>
> >> Can they comment on the applicability of the prior work suggested above?
>
> The Label super-resolution networks assume that the label is given at the low-resolution level (in the form of one label per block of the image) and known distribution of high-resolution labels is conditioned on the label of the low-resolution block. Our work also introduces a certain kind of low resolution labels, in the sense that the boundaries of the WL are noisy and inaccurate. We make use of the observation that the pixels that are in the partial mask (WL) set a strict constraint for the values of the SL, and do not assume that the distribution of the SL  given the WL is known.
>
> In practice, the annotation strategy is very different for this method compared to ours. Our WL gives more information as it includes details on separating and touching objects. This is not the case with the low-resolution level labels used by the Label super-resolution network. Therefore the mentioned prior work is not directly comparable to our approach.
>
> >> How do the rough labeling tools work on biomedical data where the objects are more heterogenous patterns
>
> Indeed, in the ice crystal and air bubble problem, the distributions of pixels are similar in the different classes, which makes this problem difficult to handle. Since our method provides accurate results on this problem, we believe the segmentation task should adapt easily to different distributions of pixels. Given the results we show on the Gland dataset (Subsection 4.2), we believe that our method should also generalize well to textured objects. At the same time, the detection should not be affected by heterogeneous patterns, but more extensive experiments would have to be conducted to validate this.
>
> >> Can this work be used for segmentation and prediction on crop data?
>
> We are not familiar with prediction problems associated to crop data. Our work could be relevant in the case of crop image datasets containing small training sets.
>
> >> It seems as if the improvement over the PL baseline (pseudo labels) is incremental? Can the authors provide error bars so the reader knows what the significance of the results is?
>
> We have added error bars to compare the two methods (see Figure 4). Thanks for the suggestion. In addition to the comparisons provided in Table 1 and Figure 4, we have demonstrated qualitative results in Figure 5, which shows that our approach gives accurate predictions on the contours of the objects, and this is one of our main objectives in the work.
>
> >> Can they give a more thorough comparison in terms of human effort? It is interesting to note that only 2 images give 0.82. Would 3 images give 0.94?
>
> We included a comparison of the supervision from different amounts of SL in Table 2. The $20\%$ SL ($4$ images), $30\%$ SL ($6$ images) and $75\%$ SL ($15$ images) give scores of $0.882$, $0.913$, $0.940$ respectively. Given 6x speedup on WL annotation time, the creation of $10\%$ SL + WL is $3$ times faster than $75\%$ SL which gives similar accuracy.
>
> >> What is the performance of MT U-net without the SL images (i.e without task-3)?
>
> The MT U-net without SL (i.e., without task 3) degenerates into the U-net with WL. In fact, according to the loss function (5) we used, the loss on task 3 is always zero provided no SL available. This contributes to nothing in backpropagation during the network training, and therefore MT U-net degenerates into the U-net with WL. We have provided the results on U-net with WL (second row of table 1), which therefore covers the results of MT U-net without task 3.
>
> >> Table-3: How well does MDUnet do with 9.4% SL data?
>
> The MDUnet is proposed in a fully supervised setting, and it is limited by the amount of SL, so is the single task U-net. We have demonstrated the relatively lower performance of the single task U-net (in Table 3, and also in Table 1), and similarly a significant performance reduction could be expected for the MDUnet if one reduces the SL from $100\%$ to only $9.4\%$.

---

> ### Author Response · Authors · 2019-11-09
> **Response to Review #2 (Part 1)**
>
>
> We thank the reviewer for the comments and feedback.
>
> >> The motivation for the specific structure of the multi-task blocks is not clear
>
> The design of the multi-task blocks is inspired by the architecture of U-net, which refines the feature maps in a sequence of increasing resolution on its expanding path. Let us go through the main structural ideas:
> - In task 1 (detection), for each object instance, only a partial mask is learned (see for instance the bottom left of Figure 3 and the middle of Figure 6). The remaining part of the object instances that is not detected in task 1 can sometimes be large - many pixels in diameter. This remaining part should be detected in task 3.
> - Each convolutional layer performs only local operations (typically on a few neighbourhood pixels).  This motivates us to add residual blocks on the lower resolution levels in order to manage the step that is required to go from task 1 to task 3 labels.
> - As the task 3 labels (segmentation) contain more information of the object boundaries than the task 1 labels (detection), we concatenate the feature maps for task 3 directly with the feature maps of task 2 (separation) in each multi-task block. This makes the boundary information of task 3 be shared with task 2.
>
> Since much fewer samples are annotated for task 3, we let task 1 and task 3 share parameters in each multi-task block in order to reduce some degrees of freedom for task 3. We also tried to apply multi-task blocks with separated parameters for task 1 and 3 (keeping the other parts of the network unchanged), but this suffers from a reduction in the segmentation accuracy.
>
> >> The object boundaries labels can be noisy (i.e s(2) can have noise). How does model deal with this?
> This is one of the challenges of learning with weak labels. In the creation of $s^{\rm (2)}$ we allow a certain amount of noise. This is practical as this makes the user’s job easier when doing the annotations.
>
> The label $s^{\rm (2)}$ only focuses on certain parts of the boundaries, and instead of determining the boundaries up to a single pixel we pick wider regions (scribbles) which are likely to contain the boundary pixels. Therefore, the labels $s^{\rm (2)}$ for training the model are already averaged over several neighbouring pixels around the boundary and as such render the trained model robust to small perturbations / noise in the labels. We do observe the capability of the network for doing so (see the right of Figure 5).
>
> >> Is it the case that every image in I_3 is completely labeled - i.e all segments/classes marked?
>
> Yes, the set $\mathcal{I}_3$ consists of images that are fully annotated. These annotations are derived by a combination of standard image processing methods using the weak labels $\mathcal{I}_1$ as a starting point and manual corrections of the so-derived full segmentation mask by a user. This process does require more effort than annotating images from $\mathcal{I}_1$ and $\mathcal{I}_2$. However, we emphasize that the set $\mathcal{I}_3$ is small (for example the size of $\mathcal{I}_3$ is 2 for dataset 1 and 8 for dataset 2).
>
> >> The assumption that s(3) is independent of s(1) and s(2) is not true. Instead of constraining the model to learn masks that respect the various types of labels, it seems they learn from each source independently. It is not clear how the sharing of parameters in the multitask block helps.
>
> - The assumption is that they are conditionally independent rather than independent. In fact, given the image $I$, $s^{\rm (3)}$ can be created independently to $s^{\rm (1)}$ and $s^{\rm (2)}$.
>
> - We trained our model for all tasks jointly. The effectiveness of transferring knowledge from the weak labels (WL)  to the segmentation is reflected by the improvement from the U-net trained on the WL or the strong labels (SL) only (cf. Table 1).
> We propose to learn the segmentation in the form of multiple tasks instead of learning a mask that is constrained by the various sources of labels, as this explicitly exposes the network to the statistical information of the WL which is not necessarily reproducible from the masks. For instance, the segmentation mask is not sufficient for predicting the touching objects without clear boundaries between them. We also show that the approach works well with the unbalanced  WL classes. In fact, no background pixels are labeled for the detection task, whereas those background pixels can be ignored if we only learn for a mask respecting the WL, given that the contrast between the background and the foreground can be poor.
>
> - Regarding the multi-task blocks, we referer the reviewer to the explanation of the motivation we gave at the beginning of the post. For task 3, overfitting can be an issue if the images that are labeled are extremely sparse. Sharing of labels between the task helps to prevent this.

---

> > ### Comment · AnonReviewer2 · 2019-11-13
> > **Acknowledging the rebuttal**
> >
> > I have read the author's response to all comments by other reviewers. My current remaining concern is -- we need to have more datasets so that the comparison with other papers is easier. The authors argue that one of their motivations to not use some of the mainstream datasets is that they are large. However the larger datasets can be pruned to make them small if they want to show their method's relevance.
> > I have updated my overall rating accordingly.

---

> > > ### Author Response · Authors · 2019-11-15
> > > **Acknowledging further comments**
> > >
> > > Thank you very much. We really appreciate the further comments and suggestions.
> > >
> > > We agree that it is possible to make the mainstream datasets smaller in order to examine the weak supervision methods’ performance in the scenario of less training data. In this context, a performance loss is to be expected with all weak supervision methods (including image level, or count level supervision that uses a few labels per image), but it would be interesting to see the quantitative results.
> > >
> > > As we mentioned in the response to review #1, the suggested datasets are not always accurate enough, given that one of our main objectives is to find a precise segmentation. Selecting samples with accurate masks in these datasets or refining their masks will require a lot of manual work for such experiments.
> > >
> > > Moreover, we are concerned with accurate methods for images with poor contrast and for which finding the object contours is crucial (as in our response to reviewer #3).  We are keen to explore other relevant datasets in such condition or create large datasets of this kind as benchmarks.

---

### Decision · Program_Chairs · 2019-12-19

**Decision:**

Reject

**Comment:**

The paper proposes an architecture for semantic instance segmentation learnable from coarse annotations and evaluates it on two microscopy image datasets, demonstrating its advantage over baseline. While the reviewers appreciate the details of the architecture, they note the lack of evaluation on any of popular datasets and the lack of comparisons with baselines that would be more close to state-of-the-art. The authors do not address this criticism convincingly. It is not clear, why e.g. the Cityscapes or VOC Pascal datasets, which both have reasonably accurate annotations, cannot be used for the validation of the idea. If the focus is on the precision of the result near the boundaries, then one can always report the error near boundaries (this is a standard thing to do). Note that the performance of the baseline models is far from saturated near boundaries (i.e. the errors are larger than mistakes of annotation).

At this stage, the paper lacks convincing evaluation and comparison with prior art. Given that this is first and foremost application paper, lacking some very novel ideas (as pointed out by e.g. Rev1), better evaluation is needed for acceptance.